# Preterm Infants on Early Solid Foods and Vitamin D Status in the First Year of Life—A Secondary Outcome Analysis of a Randomized Controlled Trial

**DOI:** 10.3390/nu14153105

**Published:** 2022-07-28

**Authors:** Margarita Thanhaeuser, Fabian Eibensteiner, Margit Kornsteiner-Krenn, Melanie Gsoellpointner, Sophia Brandstetter, Ursula Koeller, Wolfgang Huf, Mercedes Huber-Dangl, Christoph Binder, Alexandra Thajer, Bernd Jilma, Angelika Berger, Nadja Haiden

**Affiliations:** 1Department of Pediatrics and Adolescent Medicine, Comprehensive Center for Pediatrics, Medical University of Vienna, 1090 Vienna, Austria; margarita.thanhaeuser@meduniwien.ac.at (M.T.); fabian.eibensteiner@meduniwien.ac.at (F.E.); margit.kornsteiner.krenn@gmail.com (M.K.-K.); sophia.brandstetter@meduniwien.ac.at (S.B.); mercedes.huber-dangl@meduniwien.ac.at (M.H.-D.); christoph.a.binder@meduniwien.ac.at (C.B.); alexandra.thajer@meduniwien.ac.at (A.T.); angelika.berger@meduniwien.ac.at (A.B.); 2Department of Clinical Pharmacology, Medical University of Vienna, 1090 Vienna, Austria; melanie.gsoellpointner@meduniwien.ac.at (M.G.); bernd.jilma@meduniwien.ac.at (B.J.); 3Department of Laboratory Medicine, Klinik Hietzing, Wiener Gesundheitsverbund, 1130 Vienna, Austria; office.koeller@gmail.com (U.K.); wolfgang.huf@gesundheitsverbund.at (W.H.)

**Keywords:** preterm infant, solid foods, vitamin D status, vitamin D intake

## Abstract

Preterm birth places infants at high risk for mineral and micronutrient deficiencies important for bone health. The aim of this study was to examine whether two timepoints for the introduction of solid foods in preterm infants have an impact on vitamin D status in the first year of life. This is a secondary outcome analysis of a prospective, randomized trial on very low birth weight (VLBW) infants, randomized to an early (10–12th week corrected age) or a late (16–18th week corrected age) complementary-feeding group. Vitamin D status was assessed by blood samples taken at 6 weeks, 6, and 12 months corrected age. In total, 177 infants were randomized (early group: *n* = 89, late group: *n* = 88). There was a tendency toward lower levels of serum 25-OH-vitamin D in the early group throughout the first year of life (*p* = not significant (n.s.)); no differences were detected in the other parameters. At 6 months corrected age, infants of the early group had a significantly higher incidence of vitamin D deficiency. The timepoint of the introduction of solid foods had no impact on the serum 25-OH-vitamin D levels and other parameters important for bone health but showed a tendency toward lower levels in the early-feeding group.

## 1. Introduction

Preterm birth places infants at high risk for mineral and micronutrient deficiencies important for bone health, such as calcium, phosphorus, and vitamin D, as accretion of calcium and phosphorus is highest in the third trimester of pregnancy [1]. The vitamin D levels of infants at birth are 50–75% of their mother’s levels, whereas especially preterm infants show deficient vitamin D values at birth [2,3]. The values are lowest in infants born in the winter season, in infants of mothers with dark skin, who are living at higher latitude, and in those practicing complete body covering due to cultural or religious reasons [4,5,6].

Low neonatal bone mineral content and vitamin D deficiency lead to a high risk of developing rickets in preterm infants, with the risk being highest in the most preterm infants [7]. Vitamin D and mineral status are not only important for bone health but might also have an impact on respiratory morbidity as it has biological importance in lung development and surfactant synthesis [8]. Vitamin D plays a critical role in cellular function and shows other extra-skeletal anti-inflammatory and immune effects as well [9,10,11].

To meet the needs of preterm infants up to term and avoid deficiencies, current recommendations suggest a vitamin D supplementation of 400–1000 IE/day until one year corrected age and the use of a fortifier for breastfed infants or preterm infant formula [12,13]. The wide range in the supplementation recommendation derives from regional differences (European Society for Paediatric Gastroenterology Hepatology and Nutrition (ESPGHAN), 800–1000 IE/day; American Academy of Pediatrics, 400–1000 IE/day) [14]. Many studies on preterm infants with different doses of vitamin D supplementation and effects on 25-OH-vitamin D levels in the first few weeks of life exist [15,16,17], but data on the discharged preterm infant and their first year of life are limited [18]. Furthermore, recommendations on the best mode and timepoint of monitoring the vitamin D status of preterm infants are not available [12].

The influence of the timepoint of the introduction of solid food on vitamin D and bone mineral intake in preterm infants is an unexplored field so far. Moreover, recommendations on the optimal food composition for sufficient intake are missing. The data of term infants show promising positive effects on bone mineral content in infants introduced to solid foods before 4 months [19].

The aim of this study was to examine whether two different timepoints for the introduction of solid foods in preterm infants have an impact on their vitamin D levels and other parameters important for bone health in the first year of life.

## 2. Materials and Methods

This pre-specified secondary outcome analysis of a prospective, randomized, two-arm intervention trial (ClinicalTrials.gov (https://clinicaltrials.gov): NCT01809548) on preterm infants on early solid foods was conducted at a level IV neonatal care unit of the Department of Pediatrics, Division of Neonatology, Pediatric Intensive Care and Neuropediatrics (Medical University of Vienna, Austria) from October 2013 to February 2020. Primary outcome and details on study design were published previously [20].

Preterm infants with a birth weight <1500 g and a gestational age <32 weeks of gestation were included. Excluded were all infants with diseases affecting stable growth (i.e., necrotizing enterocolitis (NEC) with short bowel syndrome [21], Hirschsprung disease [22], chronic inflammatory bowel disease [23], bronchopulmonary dysplasia (BPD) [24], congenital heart disease [25], major congenital birth defects or chromosomal aberrations).

Parents were contacted at the first appointment at the neonatal outpatient clinic at their infants’ expected date of birth. After informed consent was obtained, infants were stratified (breastfed or formula fed) and randomized to an early complementary-feeding group with introduction of solid food between the 10th and 12th week of life corrected age or a late complementary-feeding group with introduction of solid food between the 16th and 18th week of life corrected age. Age-appropriate standardized complementary food in addition to breastfeeding or formula was provided until one year corrected age. Details on the standardized feeding boxes were previously published [20].

Infants with a weight <10th percentile at discharge received fortified breast milk (Aptamil FMS, Milupa Nutricia GmbH, Frankfurt, Germany) or preterm discharge formula until 52 weeks corrected age. Calcium and phosphorus supplementation was controlled by weekly monitoring of urinary mineral excretion. If urinary calcium was <2 mmol/L, supplementation with calcium-phosphorus capsules twice a day (one capsule of 105 mg calcium-glycerophosphate and 134 mg calcium-gluconate) was started until stable elevated values were reached. Vitamin D3 supplementation (400 IE/day, Oleovit D3, Fresenius Kabi Austria GmbH, Graz, Austria) was provided until one year corrected age, supplementation with iron polymaltose (2–3 mg/kg/day, Ferrum Hausmann, iron oxide polymaltose complex, Vifor France, Paris, France) until iron-rich solid foods were fed on a regular basis. Furthermore, infants received a multivitamin preparation (vitamins A, E, D3—250 IE/day, B1, B2, B6, C, Niacin, Pantothenic acid; Multibionta, Merck Selbstmedikation GmbH, Darmstadt, Germany) until one year corrected age. The total daily supplemental vitamin D intake was 650 IE, consisting of 400 IE/day from Oleovit D3 and 250 IE/day from Multibionta. Possible additional supply from infants’ formula was not taken into account.

The present study is primarily focusing on differences in 25-OH-vitamin D levels of patients throughout the first year of life between study groups assessed by serial blood samples taken before the introduction of solid foods at 6 weeks and at 6 and 12 months corrected age. Secondary outcomes included measuring serum levels of other parameters important for bone status, such as calcium, phosphorus, albumin, alkaline phosphatase, and parathyroid hormone (PTH). Furthermore, incidence of vitamin D deficiency defined as serum 25-OH-vitamin D levels <50 nmol/L was assessed at all visits [12].

25-OH-vitamin D and PTH were measured with Chemiluminescence Immunoassay (CLIA, DiaSorin Liaison, Saluggia, Italy); calcium, phosphorus, albumin, and alkaline phosphatase were all measured with a cobas 8000 modular analyzer (Roche Diagnostics International AG, Rotkreuz, Switzerland).

Patient flow chart was published recently [20]. Baseline characteristics, parameters on neonatal morbidity, and nutritional parameters are shown in Table 1.

### 2.1. Study Visits

Infants were invited to study visits at expected due date, 6 weeks, 12 weeks, 6 months, and 12 months corrected age, along with clinical routine care visits at the neonatal outpatient clinic. At every visit, anthropometric measurements were collected; results were published recently [20]. Blood samples measuring serum levels of 25-OH-vitamin D, calcium, phosphorus, albumin, alkaline phosphatase, and parathyroid hormone were taken at 6 weeks corrected age before complementary feeding was started, as well as at 6 and 12 months corrected age.

Written informed consent from one parent was sufficient due to low risk for participants. The study was approved by the Ethics Committee of the Medical University of Vienna (EK: 1744/2012) and registered at clinicaltrials.gov (NCT01809548).

### 2.2. Statistical Analysis

In general, for ordinal and nominal data, absolute and relative frequencies were calculated. Mean and standard deviation was calculated for continuous variables. Vitamin D supplementation is displayed as IE/day which was calculated as cumulative vitamin D supplementation from discharge until the corresponding visit (6 weeks corrected age, 6 months corrected age, 12 months corrected age) divided by number of days of supplementation until the corresponding visit. Differences between groups (early vs. late introduction of complementary feeding) in bone metabolism-associated blood parameters (25-OH-vitamin D, calcium, phosphorus, albumin, alkaline phosphatase, parathyroid hormone) were analyzed using a linear mixed-effects model with a random intercept to account for possible correlation between siblings of multiple births and the following fixed effects: study group, gestational age at birth, sex, nutrition at discharge (breastfed vs. formula), and vitamin D supplementation (IE/day). The models’ results are reported as estimated marginal means, respective 95% confidence intervals (95% CI), and uncorrected *p*-values.

We further evaluated study group differences over a clinically meaningful and established cut-off for vitamin D deficiency (serum 25-OH-vitamin D levels <50 nmol/L) with a mixed-effects logistic regression, fitted through study group, gestational age at birth, sex, nutrition at discharge (breastfed vs. formula), and vitamin D supplementation (IE/day). Statistical analysis was conducted using R software (R Core Team 2020, https://www.R-project.org/, accessed on 20 June 2022).

## 3. Results

### 3.1. Screening and Participants

In total, 177 infants were randomized between October 2013 and April 2020, with 89 infants to the early group and 88 infants to the late group.

### 3.2. Baseline Characteristics and Neonatal Morbidity

The detailed demographic parameters as well as information on neonatal morbidity and nutritional parameters were published previously [20]; the mean gestational age was 27 + 1 weeks in both groups, respectively. The infants had a mean birth weight of 941 g in the early group and 932 g in the late group. The parameters important for vitamin D status are shown in Table 1. There were no significant differences between the study groups.

### 3.3. Primary Outcome

The results from the primary outcome are shown in Figure 1. The 25-OH-vitamin D levels did not differ significantly between the study groups at 6 weeks, 6 months, and 12 months corrected age, respectively, but decreased over time. The 25-OH-vitamin D levels were lower in the early-feeding group at all three timepoints, but this finding was statistically not significant. Both groups showed deficient mean 25-OH-vitamin D levels throughout the first year of life, except for the late group at 6 weeks corrected age.

### 3.4. Secondary Outcomes

Table 2 shows the results from the mixed-effects model analysis. The mean supplemental vitamin D intake decreased between 6 and 12 months corrected age but stayed within the recommended range of 400–1000 IE/day. The calcium, phosphorus, albumin, parathyroid hormone, and alkaline phosphatase showed no significant differences and changes between the study groups throughout the first year of life.

At 6 weeks corrected age, 46% of the early and 51% of the late group (*p* = 0.9); at 6 months corrected age, 53% and 40% (*p* = 0.03); and at 12 months corrected age, 65% and 57% (*p* = 0.16) showed deficient Vitamin D values < 50 nmol/L, respectively. After correction for multiple testing, differences were statistically not significant.

The detailed information of the mixed-effects model on the parameter 25-OH-vitamin D are shown in Appendix A Appendix A.

## 4. Discussion

This secondary outcome analysis of a randomized controlled trial showed that the timepoint of the introduction of solid foods had no impact on vitamin D status in the first year of life in VLBW preterm infants. Regarding growth and vitamin D status, the early introduction of solid foods in VLBW infants is safe [20]. Infants did not benefit from an early start with solid foods when it comes to bone health, as solid foods do not necessarily provide higher vitamin D intake. Moreover, a later introduction of complementary feeding had no negative influence on vitamin D status. Thus, we consider that starting solids according to their neurological abilities is safe concerning bone health in very low preterm infants.

Infants from our study had a mean birth weight below 1000 g and were introduced to complementary food under standardized conditions. Our data provide a longitudinal measurement of 25-OH-vitamin D levels and other parameters important for bone health throughout the first year of life of a large cohort of preterm infants, with infants suffering from comorbidities affecting stable growth being excluded.

Overall, the mean 25-OH-vitamin D levels in both groups were far below the normal range of 80–200 nmol/L despite a supplementation between 650 and 670 IE/day, which corresponds to the recommended range from 400 to 1000 IE/day [12]. At 6 months and 12 months corrected age, infants showed deficient mean 25-OH-vitamin D values < 50 nmol/L in both groups. According to Koletzko et al., 25-OH-vitamin D levels > 75–80 nmol/L represent a normal vitamin D status. This was only reached by two infants in each group at 6 months corrected age and by none of the infants at 12 months corrected age [12].

When compared to other studies on preterm infants with different doses of vitamin D supplementation, infants reached higher 25-OH-vitamin D levels with equal or even lower supplementation [15,16,17]. All those studies reported on early supplementation and their effects on vitamin D levels in preterm infants mostly during hospital stay, but data after discharge of preterm infants are scarce.

Recently, Jung et al. reported on vitamin D status and deficiency during the first year of life of preterm infants in Korea, which is a country with a very high overall prevalence of vitamin D deficiency [18]. They collected 25-OH-vitamin D levels at birth, 6 months, and 12 months of age, and infants were supplemented with 400 IE/day of vitamin D. Still, at 6 months of age, 31% were vitamin D deficient, and at 12 months of age, only 27% of the preterm infants showed deficient 25-OH-vitamin D levels. The authors also reported on an association of higher 25-OH-vitamin D levels at birth and a lower incidence of vitamin D deficiency in the first year of life [18]. However, these infants were older and had a higher birth weight compared with our cohort.

Longitudinal measurements of 25-OH-vitamin D levels and other important factors for bone health in the first year of life of preterm infants with a mean birth weight below 1000 g are scarce. Further studies in this vulnerable group of patients are of high interest, because our findings suggest that an even higher dose of vitamin D supplementation might be necessary to meet the need of extremely low birth weight (ELBW) infants and to avoid inadequate bone mineralization. Especially the third trimester of pregnancy is important for bone mineralization, setting preterm infants at risk for the development of rickets of prematurity or metabolic bone disease [27]. The calcium, phosphorus, PTH, and alkaline phosphatase levels of the infants in our study were stable despite low vitamin D levels. This suggests that to correct a possible deficit in calcium and phosphorus levels, no bone demineralization was necessary. Moreover, no case of a fracture caused by rickets of prematurity was reported within our cohort.

Taylor et al. postulated a laboratory cut-off value for an optimal bone mineralization in preterm infants: In a study on VLBW infants, a threshold of 120 nmol/L 25-OH-vitamin D was detected at term-equivalent age [12,28], indicating that a certain level of 25-OH-vitamin D concentration is necessary to guarantee the optimal physiological function of vitamin D-dependent processes [28]. Above this concentration, bone mineralization in the femur no longer increased [28]. Comparable data of infants with a birth weight < 1000 g do not exist yet but would be of high interest.

It is known that the smaller and the more preterm infants are, the higher the incidence of metabolic bone disease is [29]. With a higher incidence of comorbidities, a higher incidence of steroid and diuretic therapy, and postnatal immobilization, the chance of bone demineralization increases [29].

We can only assume that especially in the smallest and sickest infants, the need for supplemental vitamin D is even higher as currently suggested to achieve proper 25-OH-vitamin D concentrations during the hospital stay and beyond. In addition, preterm infants should catch-up their growth deficits during the early post-discharge period and therefore need sufficient vitamin D stores [30].

What are future approaches to improve the 25-OH-vitamin D levels of extremely preterm infants? Data from the present study indicate that an increase in vitamin D supplementation and detailed recommendations for regular monitoring during the first year of life are necessary. Recent studies already show promising improvement using adjusted vitamin D supplementation [31,32]. Artman et al. went up to 800 IE/day in infants with low serum 25-OH-vitamin D levels [31]. Kołodziejczyk-Nowotarska et al. increased additional vitamin D intake up to 800–1000 IE/day if necessary and thus 93% of infants in the intervention group achieved safe vitamin D levels (50–200 nmol/L) at 52 weeks corrected age compared to 70% in the control group [32].

Moreover, parents’ awareness for a proper vitamin D intake and necessary supplementation should be raised at doctors’ visits and by nutritional counseling.

This study comes with several limitations as it is a secondary outcome analysis of a randomized controlled study and was not powered for the detection of a difference in vitamin D status between study groups. Data of bone mineralization itself are not available, which would have been of high interest as well. Blood samples were frozen and analyzed in big blocks, so no direct action could be made. Vitamin D supplementation was assessed at every visit, but not on a daily basis. Data on maternal prenatal vitamin D supplementation and serum vitamin D levels at birth were not available, so the infants’ levels could not be adjusted. Supplementation during pregnancy already was addressed by many studies [33], because pregnant women are known to have lower 25-OH-vitamin D levels compared to non-pregnant women [34] and because the infants’ vitamin D levels reach only 50–75% of their mothers [3]. Still, national recommendations on prenatal vitamin D supplementation in pregnancy are missing because of low evidence [33]. In extremely preterm infants who spend their first months of life in an intensive care ward, possibly getting different therapies with an influence on bone health such as diuretics and steroids, the question arises if a correlation of the vitamin D levels of infants and mothers can still be found.

## 5. Conclusions

The timepoint of the introduction of solid foods had no significant impact on vitamin D status in the first year of life of the VLBW preterm infants but showed a trend toward lower serum 25-OH-vitamin D levels in infants in the early complementary-feeding group without being clinically significant. As 25-OH-vitamin D values were low throughout the first year of life of the VLBW infants, recommendations for vitamin D supplementation during the first year of life in preterm infants should be reconsidered, and regular monitoring with the possibility to adjust therapy should be introduced.

## Figures and Tables

**Figure 1 nutrients-14-03105-f001:**
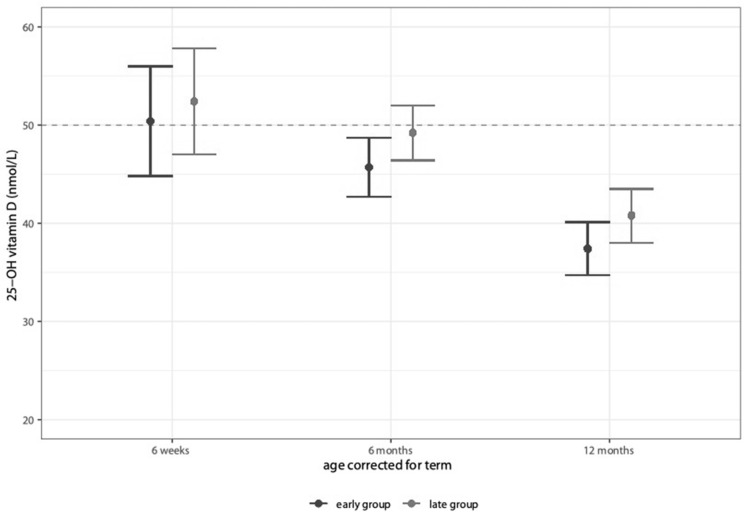
25-OH-vitamin D levels at 6 weeks, 6 months, and 12 months corrected age.

**Table 1 nutrients-14-03105-t001:** Baseline characteristics and neonatal morbidity.

Parameter	Early Group (*n* = 89)	Late Group (*n* = 88)
Obstetric and parental parameters		
Multiple pregnancy	32 (36)	28 (31.8)
Cesarean delivery	78 (87.6)	84 (95.5)
Prenatal steroids (full course)	47 (52.8)	57 (64.8)
Premature rupture of membranes	39 (43.8)	39 (44.3)
Preeclampsia	9 (10.1)	8 (9.1)
Gestational diabetes	3 (3.4)	3 (3.4)
Smoking habits		
Before pregnancy	19 (21.3)	14 (15.9)
During pregnancy	3 (3.4)	1 (1.1)
After pregnancy	1 (1.1)	2 (2.3)
Always	9 (10.1)	14 (15.9)
Age of mother at birth	32.5 [±5]	32.6 [±6.8]
Dark skin	7 (7.9)	4 (4.5)
Neonatal parameters		
Male sex	56 (62.9)	42 (47.7)
Gestational age (days)	190 [±16]–27 + 1	190 [±16]–27 + 1
Birth weight (g)	941 [±253]	932 [±256]
Small for gestational age	7 (7.9)	5 (5.7)
Born in winter	22 (24.7)	16 (18.2)
Gestational age (days) at discharge	265 [±12]–37 + 6	265 [±15]–37 + 6
Breast milk feeding at discharge	30 (33.7)	21 (23.9)
Calcium-Phosphorus supplementation		
CaPh suppl. after discharge	16 (18)	18 (20.5)
CaPh suppl. after discharge, days	12.7 [±38.6]	10.2 [±23.3]
Neonatal morbidity		
NEC grade I and II	4 (4.5)	0 (0)
PDA	34 (38.2)	33 (37.5)
ROP ≥ grade III	5 (5.6)	5 (5.7)
IVH grade I and II	9 (10.1)	4 (4.5)
IVH grade ≥ grade III	4 (4.5)	6 (6.8)
PVL	0 (0)	2 (2.3)

Categorical data are presented as numbers with percentages in round parentheses. Continuous data are presented as the mean ± standard deviation in squared parentheses. CaPh suppl.—Calcium Phosphorus supplementation, IVH—intraventricular hemorrhage, NEC—necrotizing enterocolitis, PDA—persisting ductus arteriosus, PVL—periventricular leukomalacia, ROP—retinopathy of prematurity, Small for gestational age defined as weight at birth < 10th percentile [26]).

**Table 2 nutrients-14-03105-t002:** Vitamin D status.

Parameter	6 Weeks Corrected Age		6 Months Corrected Age		12 Months Corrected Age	
	Early Group (*n* = 89)	Late Group (*n* = 88)	Early Group (*n* = 89)	Late Group (*n* = 88)	Early Group (*n* = 89)	Late Group (*n* = 88)
Vitamin D intake by supplements (IE/day)	656 (631–682)	664 (638–689)	649 (622–676)	666 (640–692)	598 (563–634)	612 (576–647)
Vitamin D status						
25-OH-vitamin D (nmol/L)	50.4 (44.8–56.0)	52.4 (47.0–57.8)	45.7 (42.7–48.7)	49.2 (46.4–52.0)	37.4 (34.7–40.1)	40.8 (38.0–43.5)
Calcium (mmol/L)	2.6 (2.5–2.7)	2.6 (2.5–2.6)	2.6 (2.6–2.6)	2.6 (2.6–2.6)	2.6 (2.5–2.6)	2.6 (2.6–2.6)
Albumin (g/dL)	3.5 (3.4–3.6)	3.4 (3.4–3.5)	4.1 (4.0–4.1)	4.1 (4.1–4.2)	4.2 (4.2–4.3)	4.2 (4.1–4.3)
Calcium corrected (mmol/L)	3.0 (2.9–3.1)	3.0 (2.9–3.1)	2.6 (2.5–2.6)	2.5 (2.5–2.6)	2.4 (2.3–2.4)	2.4 (2.4–2.5)
Phosphorus (mmol/L)	2.1 (2.1–2.2)	2.2 (2.1–2.2)	2.1 (2.0–2.1)	2.1 (2.0–2.1)	1.9 (1.9–2.0)	2.0 (1.9–2.0)
PTH (pg/mL)	35.7 (25.1–46.2)	38.5 (28.3–48.6)	29.7 (25.6–33.7)	30.5 (26.8–34.3)	29.4 (25.9–33.0)	30.7 (27.1–34.4)
AP (U/L)	310.3 (283–337)	321.9 (296–348)	234.4 (219–250)	243.1 (229–258)	293.8 (214–373)	279.0 (196–362)
Incidence of vitamin D deficiency						
Vitamin D deficiency	41/76 (54%)	45/79 (57%)	47/70 (67%) *	35/71 (49%) *	58/65 (89%)	50/62 (81%)

Data from a mixed-effects model are shown and are presented as the estimated marginal mean and 95% confidence intervals (CI) in parentheses. Vitamin D deficiency is presented as number of patients and percentage in parentheses. Calcium corrected was calculated as follows: Calcium + (0.8 × (4-Albumin)). *p* values < 0.05 were considered statistically significant; parameters with significant differences before correction for multiple testing (Bonferroni) were marked with *. After correction for multiple testing (Bonferroni), no significant differences were detected. PTH, parathyroid hormone; AP, alkaline phosphatase.

## Data Availability

The study protocol and the individual participant data that underlie the results reported in this article, after de-identification, are available upon request from the corresponding author 6 months after publication. Researchers will need to state the aims of any analyses and provide a methodologically sound proposal. Proposals should be directed to nadja.haiden@meduniwien.ac.at. Data requestors will need to sign a data access agreement and, in keeping with patient consent for secondary use, obtain ethical approval for any new analyses.

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
