# Peer review of "Preterm Infants on Early Solid Foods and Vitamin D Status in the First Year of Life—A Secondary Outcome Analysis of a Randomized Controlled Trial"

_nutrients, 2022, doi:10.3390/nu14153105_

Round 1
Reviewer 1 Report
The paper reports the impact of two different strategies for introducing solid foods in preterm infants, on serological markers for bone health in the first year of life.
The title is of the manuscript is descriptive, and focuses on the main and most novel findings.
The abstract provides a concise overview of the study, however requires some minor language editing.
The introduction section requires some improvement. I feel it would be necessary to add more information on vitamin D status in ex-preterm infants in the first year of life with up - to date references. Please elaborate on potential limitations, controversies or knowledge gaps of those studies.
The methods section reveals all necessary information regarding samples and methods. The methodology has been described in sufficient detail, with clearly defined outcomes. It seems that the study was conducted objectively and without bias. Sufficient ethical approvals have been obtain by the study group.
Minor comments
The results ar clearly presented, however can the authors please provide more detail on how Vitamin D intake supplements were calculated. Maybe I am unaware of the vitamin D formula used in Germany? Furthermore, why do the majority of infants have daily intakes of around 600, when the intake range is 400-1000IU/d)?
The discussion section should provide more information on the hypothetical potential of early introduction of solids to increase vitamin d levels. It would be interesting know the authors explanation on why does early introduction fail to increase vitamin D, as I feel the research question has not been answered fully.
Looking forward to receiving the revised manuscript.
Author Response
Please see our answers in the file attached.

Reviewer 2 Report
Thank you for the opportunity to review this manuscript. It addresses an important topic with sparse data in the literature about the optimal management of Vit D supplementation and introduction of solid food in VP infants.
However, I think the manuscript should be further improved.
Introduction:
L44: Just a minor issue, please put fortifier for breastfed infants first and then preterm formula as we strongly recommend breastmilk for all preterm infants due to the documented benefits.
Material and Methods (perhaps “Patients and Methods”?):
Please include a flow chart as in reference No.15 or at least mention that a flow chart can be found there. Please specify whether Vit D intake in formula was calculated.
Did you asses how frequent Vit D and the multi-vitamin was actually given (or forgotten?), did you assess adherence in any way? Was there any consequence for low Vit D levels, i.e. did these infants get a recommendation to increase Vit D intake?
Legend to table 1: please correct spelling of phosphorus, please delete PRBC as I can’t find it in the table. Please specify which percentiles were used to classify SGA.
Table 2: please specify in the legend that results of mixed effect models are shown. Please include in a supplement at least for 25-OH Vit D the detailed results of the variables included in the model.
Discussion:
It is a bit difficult to understand primary and secondary outcomes of this analysis if you mention the primary outcome of the initial study in the first paragraph. Please rephrase. The differences seen in Vit D levels are all non-significant. To emphasise that Vit D deficiency is significantly more frequent in one group is statistically correct, but does it really matter clinically if the other group is just a tiny bit above the cut-off? The authors used mixed effect models, did they identify risk groups who should be monitored for severe Vit D deficiency more closely? Did those with high Vit D supplementation (1000IE/d) profit? This dataset is in my opinion very important for further recommendations and studies with possibly higher Vit D supplementation.
As the authors mention in term born infants there is a concern about Vit D deficiency in those with later introduction of solid food. In preterm infants there is a large variability of corrected age at introduction of solid food with some parents introducing it at a chronological age of 3-4 month, i.e. sometimes corresponding to a corrected age of just about term equivalent (doi:10.1038/ejcn.2015.54). It is very reassuring that later introduction has no negative effect on Vit D status and parents ca be reassured to wait until the children show interest in solid foods and have the oral motor ability. I miss this message in the discussion. I do not really see a benefit in the present supplemental material. As I understood those are results of a different study group. There is a reference where these data can be found, why have them in this manuscript? However, as I mentioned I would like to see the mixed effect models in detail in the supplement.
Reviewer 3 Report
It is a well-constructed article, with adequate methodology and relevant results.
It is a very interesting article that portrays a randomized study on the introduction of early solid foods and vitamin D status in preterm infants.
